# A Simple Repair Algorithm for Descemet’s Membrane Detachment Performed at the Slit Lamp

**DOI:** 10.3390/jcm11237001

**Published:** 2022-11-27

**Authors:** Fan Li, Zhe Zhu, Lubo Fan, Gengrong Yi, Xiaowei Zhu, Naiyang Li

**Affiliations:** 1Eye Center, Zhongshan City People’s Hospital, Zhongshan 528403, China; 2Department of Medicine, Division of Regenerative Medicine, School of Medicine, University of California San Diego, La Jolla, CA 92093, USA

**Keywords:** Descemet’s membrane detachment, corneal edema, phacoemulsification, Pentacam, anterior segment optical coherence tomography, slit lamp

## Abstract

Our study aims to investigate a simple repair algorithm for Descemet’s membrane detachment (DMD) following phacoemulsification with Pentacam and anterior segment optical coherence tomography (AS-OCT). Twelve patients with DMD were included in this retrospective study. All cases had persistent corneal edema after phacoemulsification and no improved response to conservative treatment. The repair algorithm consisted of delineating the DMD with the Pentacam and AS-OCT, paracentesis, and intracameral air bubble performed at the slit lamp, followed by immediate supine position. At one month, the final status of Descemet’s membrane (DM), best-corrected visual acuity, and incidence of complications were noted. DMD was involved in the visual axis in all cases. The mean interval between phacoemulsification and repair was 5.3 ± 1.2 days. Complete reattachment of DM and corneal clarity occurred in all 12 eyes. Eleven (91.7%) eyes underwent one repair procedure, while one eye (8.3%) underwent a repeat procedure. No adverse events were found. Minor post-intervention complications included temporary increased intraocular pressure due to pupillary block in one eye (8.3%). In conclusion, our modified and simplified repair algorithm for DMD can be performed safely as an outpatient procedure, with accurate delineation using a Pentacam and AS-OCT. It may provide new insight into the timely diagnosis, evaluation, and management of DMD.

## 1. Introduction

Descemet’s membrane detachment (DMD) is one of the most common sight-threatening complications of cataract surgery. It is a significant cause of corneal edema, leading to irreversible corneal endothelial decompensation in severe cases. When DMD is extensive, spontaneous resolution is unlikely because of the compromised endothelial pump function. Prompt and effective management of DMD could lead to a better visual outcome [1,2] and patient satisfaction.

Surgical treatment approaches for severe DMD include viscoelastics injection, air/gas descemetopexy, suture transfixation of Descemet’s membrane (DM), or even corneal transplantation. Consensus of intervention strategies and timing of treatment remains unclear. Viscoelastics, iso-expansile gases (sulfur hexafluoride or perfluorpropane) may increase the risk of higher intraocular pressure (IOP) or even central retinal artery occlusion. Intracameral air bubble is a practical alternative for DMD management, with reduced risk of pupillary block. Researchers have demonstrated that 100% air descemetopexy is more effective than iso-expansile gases, with 95% success in the reattachment of DM and statistical improvement in visual acuity [3]. Nonetheless, rarely, central retinal artery occlusion occurs after full-chamber air bubble injection [4]. To reduce the adverse effects following air descemetopexy, partial air release was applied after complete air filling for a certain period [5,6]. The whole procedure was time-consuming and required repeated surgeries under the microscope in the operation theatre. A modified, easily reproducible, and safe procedure is desirable in a clinical setting.

The safety profile of anterior chamber (AC) paracentesis for uveitis diagnosis [7] and Descemet’s membrane endothelial keratoplasty (DMEK) graft detachment [8,9] performed at the slit lamp have been addressed previously. The advantage of AC paracentesis is that it is safe and relatively quick as an outpatient procedure. Whether DMD can also be managed by descemetopexy at the slit lamp has not been reported. The key points of AC air injection for DMD include carefully selecting the transparent peripheral incision site to avoid further DMD. In this case, Pentacam and AS-OCT are considered valuable tools for DMD evaluation when the scope of DMD cannot be easily detected by slit-lamp microscopy due to severe corneal edema. With the guidance of the Pentacam and AS-OCT, timely management of DMD is feasible.

We hypothesized that a simple repair algorithm with an initial partial fill of air under the slit lamp guided by a Pentacam and AS-OCT would serve as a novel and attractive alternative approach for the treatment of DMD. We report a series of cases with complete resolution of DMD achieved using this Pentacam/AS-OCT-guided algorithm for management of DMD following phacoemulsification.

## 2. Materials and Methods

This monocenter, retrospective, interventional case series was conducted at the Eye Center of Zhongshan City People’s Hospital from August 2013 to August 2017. All cases with more than one quadrant of persistent corneal edema involving the visual axis for at least three days after phacoemulsification and AS-OCT-confirmed DMD were enrolled in our study. The study was approved by the Clinical Research and Animal Experiment Ethics Committee of Zhongshan City People’s Hospital (ethical approval code: 2013-06). Written informed consent was obtained from all patients.

Demographic and clinical data were documented, including (1) age; (2) gender; (3) severity of DMD; (4) interval between phacoemulsification and descemetopexy; (5) best-corrected visual acuity (BCVA); (5) endothelial cell density preoperative and one-month postoperative; and (6) complications.

Pentacam (Oculus Inc., Welzlar, Germany) measurements were performed to visualize the thicker and thinner areas of the cornea. Then AS-OCT (Topcon 3D OCT 1000 Mark II, Tokyo, Japan) scans were performed in all cases. DMD was diagnosed according to clinical manifestations and further classified as planar or non-planar, with or without scrolled edges, and central or peripheral based on AS-OCT findings. As stated previously, the severity of the DMD was classified as mild if it involved <25% of the cornea and was peripheral, moderate if it involved 25–50% of the cornea and was peripheral, and severe if it involved >50% of the cornea or involved the central cornea [3]. Scores of 1–3 were given, respectively.

The detached DM was carefully examined using the slit lamp, Pentacam, and AS-OCT preoperatively. A map of the DMD was illustrated with the detachment zone and the non-detachment area delineated. The injection site in the non-detachment area was precisely marked preoperatively with the assistance of the Pentacam and AS-OCT.

Air tamponade was performed at the split lamp in an examination room with all aseptic precautions. The patients’ intrapalpebral area was sterilized with drops of povidone-iodine solution. Patients sat in front of the slit lamp following topical anesthesia with proparacaine HCl (Alcaine^®^) ophthalmic solution (Alcon Lab., Inc., Fort Worth, TX, USA). A speculum was used to keep the eye open and a sterile cotton bud was applied to stabilize the eye. The entry site was in the area where DM was attached, as shown in the AS-OCT images. The major procedure of air tamponade consists of four steps. Step 1, a 30-gauge syringe needle was inserted from the limbus area into the anterior chamber (Figure 1A). Step 2, the syringe was retracted, and approximately 150 µL of aqueous humor was suctioned subsequently (Figure 1B). Step 3, a new sterile syringe prefilled with air was inserted from the same needle entry site, and the aseptic air was injected to fill up to approximately two-thirds of the total anterior chamber volume (Figure 1C). The last step involved pulling the syringe needle from the cornea (Figure 1D). The procedure was completed by the main operator (NL) and assistant (GY). Ten minutes after the procedure, IOP was measured by no-contact tonometer (Topcon, Japan). The patient was asked to stay supine for two hours and then as much as reasonably possible to allow the air bubble to press DM to the posterior corneal stroma. All patients had the importance of a supine position during the postoperative stage explained to them. All patients received a standard postoperative regimen consisting of prednisolone acetate 1% eye drops six times a day and topical ofloxacin 0.3% eye drops four times a day for one week. After a week, the topical antibiotic was stopped and corticosteroid eye drops were tapered over the subsequent month.

Parameters evaluated were visual acuity, IOP, corneal clarity, and DMD on slit-lamp examination. DMD status was observed after the absorption of the air bubble using slit-lamp examination and AS-OCT. Associated complications were also recorded.

Normality of data distribution was assessed by the 1-sample Kolmogorov–Smirnov test. Descriptive parameters were recorded in the form of mean and standard deviation. A paired *t*-test was applied to assess the significance of the improvement in the BCVA after the surgical procedure. *p* < 0.05 was considered statistically significant. All statistical analysis was carried out using SPSS statistical software (version17.0, SPSS, Inc., Chicago, IL, USA).

## 3. Results

Of over 5000 patients who underwent phacoemulsification, 12 eyes of 12 patients with severe DMD were included in this study. Clear corneal incision was performed at a supertemporal site for all the right eyes and a superonasal site for all the left eyes, respectively. Demographic and clinical data are summarized in Table 1. The mean age of the patients was 72.3 ± 5.3 (range: 66–84) years. Male-to-female ratio was 7:5. The mean interval between phacoemulsification and descemetopexy procedures was 5.3 ± 1.2 (range: 3–7, median: 5) days. Mean score of severity of DMD was 2.5 ± 0.5 (range: 2–3, median: 2.25). All cases of DMD had a planar (≤1 mm separation between DM and stroma) configuration. There were eight superior and four extensive DMDs. The mean pretreatment corneal thickness was 753 ± 123 μm.

Descemetopexy using air could be performed successfully at the slit lamp in all cases. All patients showed resolution of the DMD with a mean time for resolution of 16.0 ± 7.1 days (6–32 days) (Figure 2, Figure 3 and Figure 4). No corneas decompensated during the 12-month follow-up. The average corneal thickness at 4 weeks was 554 ± 83 μm. The mean pre-descemetopexy BCVA was 1.19 ± 0.28 logMAR scale (logarithm of the minimum angle of resolution), and the mean post-descemetopexy BCVA was 0.15 ± 0.11 (logMAR scale, *p* < 0.001). Mean time for the air injection procedure was 23.2 ± 8.7 s, and the air remained in the anterior chamber for 5 ± 0.9 days. Preoperative endothelial cell density decreased from 2708.6 ± 231.6 cells/mm^2^ to 1884.7 ± 190.7 one month post-descemetopexy (*p* = 0.002).

Incidence of complications following anterior chamber descemetopexy at the slit lamp is recorded in Table 1. IOP increased temporarily in one eye (8.3%) due to pupillary block, which was managed with dilating drops and anti-glaucoma drug therapy. None of these eyes showed persistent increase in IOP. Migration of the bubble into the vitreous cavity occurred in one eye, which led to the initial failure of DMD resolution and required a repeat descemetopexy (case 11). Complete resolution of DMD and comparable visual acuity were achieved after the repeat procedure without any other adverse events in this case.

## 4. Discussion

In the present study, we introduced a simple repair algorithm for DMD management which can be performed at the slit lamp in an outpatient clinical setting. The preliminary findings indicate that our algorithm is a safe and effective procedure with minimal complications. Compared to previous studies, this repair algorithm is more straightforward, quicker, and easier to operate [10,11]. Twelve cases of DMD achieved anatomic reattachment and corneal transparency. The mean time of resolution of corneal edema was 6 ± 2 days. Our results are comparable to previous studies regarding DM attachment, visual acuity improvement, and resolution of corneal edema.

This repair algorithm was more simplified than and equally effective to previous studies. It was performed using a 30-gauge syringe at the slit lamp without a side port blade [9], allowing a minimally invasive incision with probably faster recovery time. Furthermore, no partial air release was needed in our patients to avoid severe complications such as central retinal artery occlusion compared with previous studies [4]. Additionally, drainage of predescemetic fluid by stab incision on the cornea was not necessary to achieve DM attachment as previously performed in other studies [10], making this procedure quicker and easier.

The repair algorithm for DMD achieved a relatively high success rate and low complication rate. Key factors contributing to the satisfactory outcome of this procedure were as follows: (1) Prompt recognition and accurate positioning of the DMD region. Generally, the transparent area of the cornea is the region where DM was attached [1]. Prior to the repair procedure, we initially evaluated the DMD region under the slit lamp carefully. We then evaluated the Pentacam images to delineate the thinner area where DM was most likely to be attached, and finally precisely positioned and delineated the detachment and the non-detachment zone aided by AS-OCT. To avoid iatrogenic exacerbation of the DMD, the needle entry site was in the area where DM was attached. Previous studies show that a better prognosis of DMD relies on earlier diagnosis and management; delayed intervention is associated with poor prognosis [3,12]. Therefore, our study chose a relatively short interval between phacoemulsification and descemetopexy procedures, with a mean interval of 5.3± 1.2 days. Additionally, as the most severe DMD region in this study was exclusively within the upper right quadrant due to the superior phaco incisions, it was relatively easy for the clinician to handle the syringe and point it superiorly to allow the injection of the air at the sitting position. The patients were instructed to look straight ahead, with complete understanding and good cooperation during the procedure. (2) Partial liquid–air exchange reduced the risk of pupillary block and IOP increase. Aqueous humor was aspirated first to provide a space for air filling. Then two-thirds of the total anterior chamber volume of air was introduced, instead of the complete anterior chamber in the previous study [2]. Air tamponade is as effective as iso-expansile gases [13,14] and has the advantage of less risk of pupillary block [3]. Compared with longer-acting gas mixtures, air is safer because of rapid systemic absorption. (3) A sitting position at the slit lamp allows for improved visualization of air bubble injection into the anterior chamber compared to a supine position under the surgical microscope. The immediate supine position allows the air bubble to press DM to the posterior corneal stroma. Lying face-up is similar to the position after corneal endothelial transplantation. In our study, migration of the bubble into the vitreous cavity occurred in one eye of case 11, partly due to improper post-intervention position and zonular weakness or dialysis.

Delineation of the morphology and location of the DMD were critical for DM reattachment. The detached DM was carefully examined using the slit lamp, Pentacam, and AS-OCT preoperatively in this study. All cases were referred to us for blurred vision and corneal edema. In the presence of corneal edema, it is usually hard to diagnose DMD immediately at the slit lamp [15]. The Pentacam allows us to view the corneal thickness to decide which area of DM is still attached. AS-OCT provides high-resolution cross-sectional images of the anterior segment, which are quite helpful for diagnosing and monitoring various anterior segment disorders [15,16]. AS-OCT can scan through a relatively opaque cornea, and minimal experience is required for image acquisition, making it an indispensable tool in corneal disorders, especially in cases of corneal edema in the postoperative period, during which a definitive diagnosis is required to plan on appropriate management. When AS-OCT was performed, DMD was definitely diagnosed in these cases. In addition, AS-OCT can determine the configuration (planar or scrolled) and location of DMD, which are vital for planning the surgical technique. Using AS-OCT, we could successfully classify types of detachment. Therefore, AS-OCT allows for an early diagnosis and treatment, thus significantly reducing the time of recovery. It also allows for the selection of cases that can be treated conservatively. Using AS-OCT images, the morphology and location of the DMD were delineated precisely [2].

DMD can lead to significant visual impairment from corneal edema. Persistent corneal edema may lead to corneal decompensation and permanent vision loss [10,17]. The incidence of significant sight-threatening DMD after phacoemulsification was 0.044% to 0.52% [14,15]. Improper operation of instrument entry into the anterior chamber is the most significant cause of DMD following phacoemulsification [18]. Although postoperative visual acuity and corneal thickness returned to normal levels afterward, the corneal endothelial cell density decreased by 30% from preoperative density in our study. The endothelial cell loss rates at similar time intervals after phacoemulsification in patients whose surgery was not complicated by DMD were reported to range from 7% to 15%, with variation being affected by axial length, depth of anterior chamber, and femtosecond laser platforms versus standard phacoemulsification, pupillary dilatation size, and phacoemulsification time [19,20,21,22]. Nonetheless, the overall endothelial cell loss rates were markedly lower in previous studies without significant DMD compared with our study with severe DMD.

Therefore, it makes more sense to prevent DMD at an early stage to maintain endothelial cell density and achieve a satisfactory long-term clinical outcome.

The time interval from cataract surgery to descemetopexy directly impacts the outcome of the descemetopexy [23]. DMD should be suspected when localized corneal edema with a distinct demarcation between the edematous and compact cornea appears. In our series, when extensive corneal edema obscured the view of the DMD, the early diagnosis was facilitated by AS-OCT. It is beneficial for us to improve the accuracy of the diagnosis and outline the prognosis of DMD [15,16].

The results of this study should be evaluated within the context of its limitations. The first limitation is the small number of participants, as only patients with severe cases of DMD after phacoemulsification were included. A larger cohort should be included for further safety investigation. The second limitation is that most patients enrolled in this study had superior DMD due to the location of the main incision site during cataract surgery in our center. Thus, it was relatively easy for the clinician to perform this procedure at the sitting position at the slit lamp and all cases were cured successfully with air. Injecting in this particular position might be challenging when the Descemet’s membrane detachment involves the inferior cornea. Therefore, different types of incisions with DMD in other locations should be compared in the future. Moreover, it should be noted that this study was not designed to compare this new DMD repair algorithm performed at the slit lamp versus traditional descemetopexy performed in the surgical theater, but rather to provide an alternative and simple approach for clinicians when the operating room is not available and the waiting list of patients is long.

## 5. Conclusions

We provide an effective and straightforward approach for DMD management following phacoemulsification, which consists of delineation guided by AS-OCT and Pentacam, paracentesis, and partial air bubble at the slit lamp, followed by supine position.

## Figures and Tables

**Figure 1 jcm-11-07001-f001:**
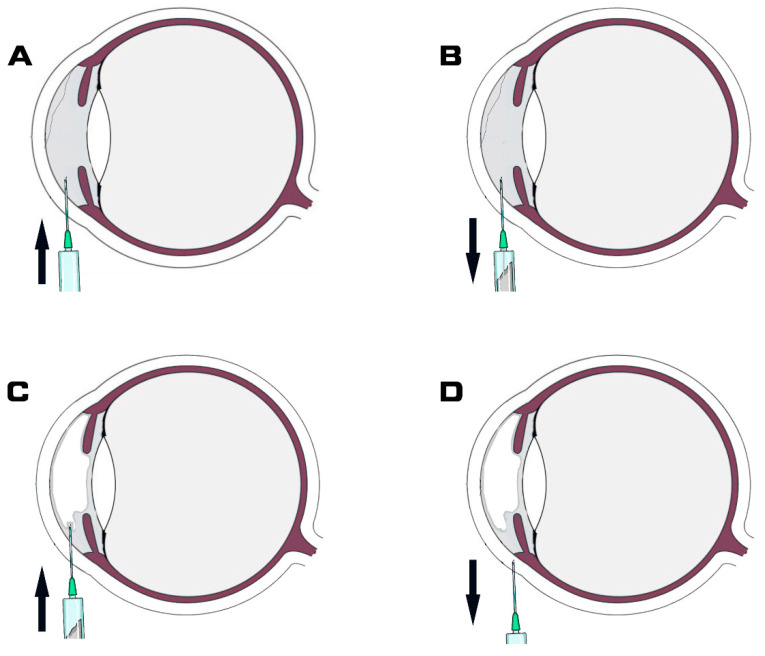
Schematic illustration of intracameral air bubble injection. (**A**) Syringe needle was inserted from limbus area into anterior chamber. (**B**) Syringe was retracted and subsequently aqueous humor was suctioned. Approximately 150 µL of aqueous humor outflow into syringe. (**C**) Air was injected into the anterior chamber until there was approximately two-thirds air fill of the total anterior chamber volume. (**D**) Pulling the syringe needle from cornea. Arrow represent the direction of the needle.

**Figure 2 jcm-11-07001-f002:**
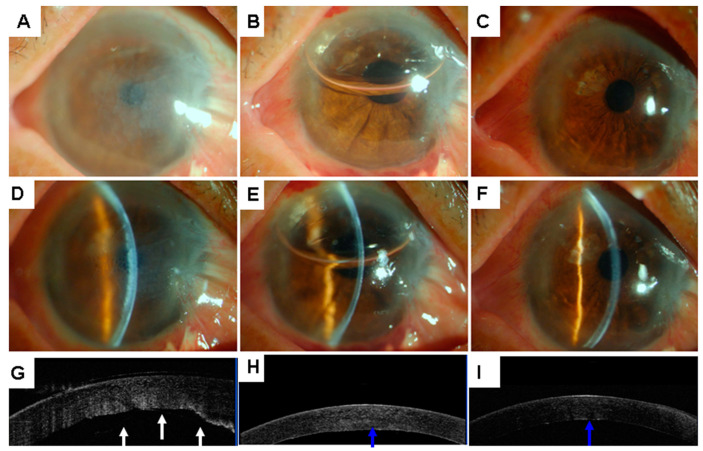
Slit-lamp photographs (**A**–**F**) and AS-OCT images (**G**–**I**) of case 1. Slit-lamp view of diffused corneal edema (**A**,**D**) and detached Descemet’s membrane (**G**, white arrow) with stromal edema on day 5 after primary phacoemulsification. Slit-lamp view one day after air injection showing clear cornea (**B**,**E**) and reattachment of the Descemet’s membrane (**H**, blue arrow). Slit-lamp view one month later showing clear cornea (**C**,**F**) and reattachment of Descemet’s membrane (**I**, blue arrow). AS-OCT = anterior segment optical coherence tomography.

**Figure 3 jcm-11-07001-f003:**
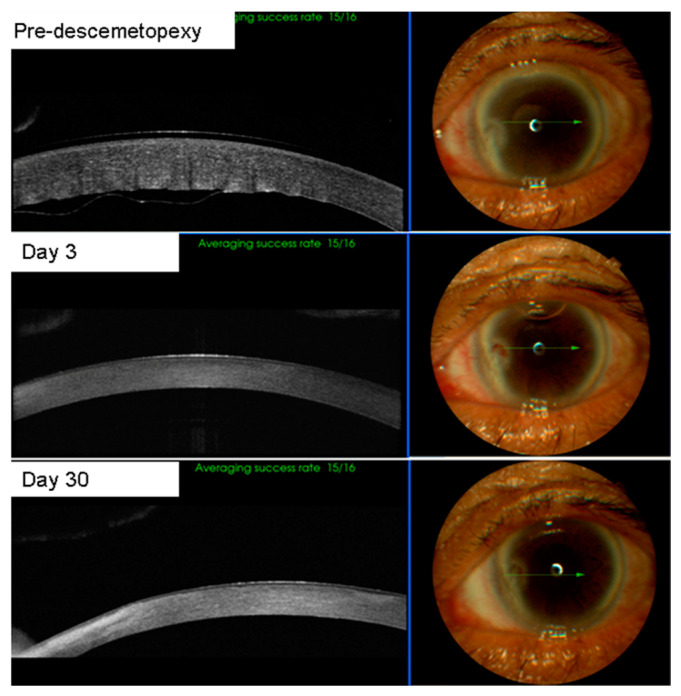
Clinical course of descemetopexy in case 3. The patient received a repair algorithm procedure and Descemet’s membrane (DM) was successfully reattached.

**Figure 4 jcm-11-07001-f004:**
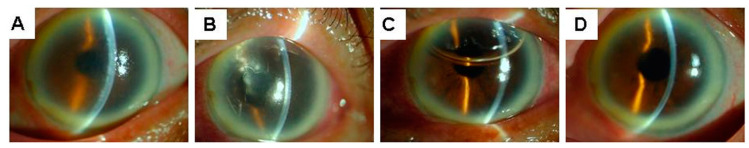
Representative clinical course of descemetopexy. Severe corneal edema (**A**) was found after phacoemulsification, and Descemet’s membrane detachment (**B**) was detected near the main corneal incision. 50% of the anterior chamber was filled with air (**C**) two days post-descemetope and cornea regained its transparency (**D**) two weeks post-descemetopexy.

**Table 1 jcm-11-07001-t001:** Patient Demographics and Characteristics.

Demographics and Characteristics	Finding *
Age, mean (SD), y	72.3 (5.3)
Sex	
Male	7 (58.3)
Female	5 (41.7)
Systemic conditions	
Diabetes	2 (16.7)
Ocular conditions	
Glaucoma	2 (16.7)
Pterygium	1 (8.3)
Complications	
Increased IOP	1 (8.3)
Repeat injection	1 (8.3)
Endothelial cell, mean (SD)	
Before cataract surgery	2708.6 (231.6)
One month post-descemetopexy	1887.7 (190.7)
Corneal thickness, mean (SD), μm	
Pre-descemetopexy	753 (12.3)
One month post-descemetopexy	554 (83)
logMAR BCVA, mean (SD)	
Pre-descemetopexy	1.19 (0.28)
One month post-descemetopexy	0.15 (0.11)

Abbreviation: logMAR, logarithm of the minimum angle of resolution; BCVA, best-corrected visual acuity; IOP, intraocular pressure. * Data are presented as number (percentage) of patients unless otherwise stated.

## Data Availability

The original contributions presented in the study are included in the article; further inquiries can be directed to the corresponding author upon reasonable request.

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
