# Peer review of "A Simple Repair Algorithm for Descemet’s Membrane Detachment Performed at the Slit Lamp"

_jcm, 2022, doi:10.3390/jcm11237001_

Round 1
Reviewer 1 Report (New Reviewer)
The topic of this study is timely and will of interest to the readers of the journal. Some images and tables were utilized in order to make the content of the text clearer. Τhe statistical analysis used was appropriate. However, some corrections are suggested to be made.
Corrections of major imprtance
-Power analysis should be performed to determine the adequacy of the sample.
-The authors should mention in detail the comparison of the results of the corresponding study and other existing studies in the discussion section.
Corrections with minor importance
-p5/ 2nd paragraph: Please, replace the term "endothelia" with the term "endothelial".
-p5/ last paragraph: Please, replace the term "Moreove" with the term "Moreover".
Author Response
Please see the attachment.

Reviewer 2 Report (New Reviewer)
Methods:
Please specify which AS-OCT device was used, the model, and the manufacturer. Please also mention which Pentacam model was used.
Further elaboration on the technique and its limitations would be helpful. The same syringe was used for both aspirating anterior chamber fluid and for injecting air, this would require the positioning of the syringe pointing superiorly to allow the injection of the air. Injecting in this particular position might be a challenge when the Descemet detachment involves the inferior cornea.
Statistics: a paired t-test rather than a t-test would be required to compare visual acuity before and after the procedure.
Results:
It would be desirable to report the endothelial cell loss rate in a cohort of patients whose surgery was not complicated by Descemet Membrane detachment
Author Response
Please see the attachment

This manuscript is a resubmission of an earlier submission. The following is a list of the peer review reports and author responses from that submission.
Round 1
Reviewer 1 Report
REVIEW – A simple repair algorithm for Descemet membrane detachment performed at the slit lamp
The authors have submitted an article presenting a new clinical algorithm for Descemet membrane detachment following cataract surgery on the slit lamp to the Journal of Clinical Medicine (vision section). The manuscript consists of an abstract, five keywords, the sections introduction, materials and methods, results, discussion as well as 1 table, 3 figures and 17 references.
The authors have advocated to perform Descemet detachment repair as a slit lamp procedure in order to facilitate complication management and reduce time to intervention. While these are reasonable drivers to promote change of clinical practice, I must ask the authors if the correct research question is identified to justify converting a procedure, which traditionally is performed in a theatre setting, to the slit lamp.
An intraocular surgical procedure such as Descemet detachment repair is invasive and bears certain risks such as infections, creation of another complication while performing the repair procedure or patient related problems such as anxiety or movements. It should not be discarded that certain ophthalmic procedures traditionally conducted in theatre may be performed under the slit lamp, however, the task of a researcher advocating for such a change should be to demonstrate non-inferiority of the slit lamp surgery compared to the intraoperative procedure in addition to the advantages which the change in surgical practice will offer. There is no doubt that, from a technical perspective, many procedures conducted in a theatre setting could also be performed under the slit lamp in an outpatient setting. However, the operation room provides many advantages such as enhanced sterility, more flexibility in patient ergonomics, a surgical microscope and a theatre team which generally justify increased costs and time investments.
It may not be very easy to demonstrate non-inferiority of surgical safety and outcome in a study or review because there are many context specific factors to be taken into account such as the healthcare system, the resource context, size of the treatment centre, experience of the surgeon and surgical team, patient numbers and waiting lists which will set challenges for aking generalisable recommendations on the surgical setting of intraocular procedures. A meta-analysis may be an initial first step to assess different variables and determinants of surgical safety. Importantly, research should focus on identifying the different factors which make a setting more suitable for a given surgical procedure. This is a broader task than describing a novel algorithm or procedure which of course is different when performed at a slit lamp compared to the operating room.
It could send out wrong signals to clinicians who may work in a setting which is very different from the author’s and stimulate them to convert to slit lamp based procedures which may make surgery unsafe and put patients at risk.